# Soft Tissue Masses of the Hand: A Review of Clinical Presentation and Imaging Features

**Samuel AbuMoussa** [1], **Mona Pari Roshan** [2], **Felipe Ferreira Souza** [3], **Dane Daley** [4], **Andrew Rosenberg** [5], **Juan Pretell** [6], **Natalia Fullerton** [6] and **Ty Subhawong** [3,*]

[1] Department of Radiology, Loma Linda University Medical Center, Loma Linda, CA 92354, USA
[2] Herbert Wertheim College of Medicine, Florida International University, Miami, FL 33199, USA
[3] Department of Radiology, University of Miami Miller School of Medicine, Miami, FL 33136, USA
[4] Department of Orthopaedic Surgery, Medical University of South Carolina, Charleston, SC 29425, USA
[5] Department of Pathology, University of Miami Miller School of Medicine, Miami, FL 33136, USA
[6] Department of Orthopaedic Surgery, University of Miami Miller School of Medicine, Miami, FL 33136, USA
* Correspondence: tsubhawong@med.miami.edu; Tel.: +1-(305)-585-7500

**Abstract:** More than 15% of all soft-tissue tumors arise in the hand. Because of the location of these tumors, even small abnormalities may be alarming to patients on presentation. Although the majority of lesions are benign and can be diagnosed solely by history and physical examination, additional imaging workup may be required to confirm a diagnosis or define anatomic extent of involvement. This paper aims to review the basic epidemiology, clinical presentation, imaging findings, and treatment options of the more common soft-tissue tumors of the hand.

**Keywords:** tumor; sarcoma; malignancy; neoplasms; hand; radiology

## 1. Introduction

Of all soft-tissue tumors, more than 15%, surprisingly, arise in the hands, which comprise a total of only 2% of total body surface area [1]. Soft-tissue structures such as muscle, fat, tendons, blood vessels, and nerves derive from mesodermal mesenchyme throughout the body. Due to the superficial location, most hand tumors present early and can be easily visualized and palpated upon physical examination. Although 95% of soft-tissue tumors of the hand excluding the skin are benign and thus have a good prognosis, patients tend to be alarmed by even small abnormalities [2]. Malignant tumors are rare yet reflect 2% of all hand lesions and have unique characteristics.

Despite the ability to diagnose many of these lesions solely by history and physical examination, additional imaging workup may be required to confirm a diagnosis or define anatomic extent of involvement. Thus, radiologists serve an important role in aiding clinicians in the appropriate management of these conditions. This paper provides an overview of the basic epidemiology, clinical presentation, imaging findings, and treatment options of the more common soft-tissue tumors of the hand.

## 2. Benign Masses

### 2.1. Ganglion Cysts

#### 2.1.1. Clinical Features

Of the soft-tissue tumors commonly found in the hand, the most frequently observed are ganglion cysts, which account for around 65% of all masses in the wrist and hand [3]. Although there is no clear explanation, females are affected at a rate almost three times as often as men [4]. The most common location of ganglion cysts is the dorsal wrist (70%), with the volar wrist (~20%) and volar retinacular cysts (~10%) making up the majority of the remaining common locations.

Although mostly idiopathic in nature, a recent study demonstrated that at least 10% of patients report a preceding traumatic event before the appearance of a ganglion [5]. This coincides with popular agreement that repetitive stretching of the capsular and ligamentous structures around joints stimulates production of tissue hyaluronic acid by nearby fibroblasts [6]. Small rents in the communicating joint capsule permit accumulation of synovial fluid within the ganglion. Dorsal wrist ganglion cysts usually originate from the scapholunate joint (Figure 1) while volar-based ganglion cysts originate from the radioscaphoid–scapholunate interval but can arise from the scaphotrapezial joint as well. Retinacular cysts along the finger flexor tendon sheaths most often arise at the A1 and A2 pulley [7] (Figure 2). Ganglion and synovial cysts are histologically distinct but radiologically indistinguishable and clinically treated equivalently. The myxoid degeneration of the connective tissue of the joint capsule leads to the formation of the ganglion cysts, which contain gelatinous fluid, including hyaluronic acid and mucopolysaccharides. They also have a fibrous lining [7–9], in contrast to synovial cysts that are lined with cuboidal epithelium and contain synovial fluid [7,8].

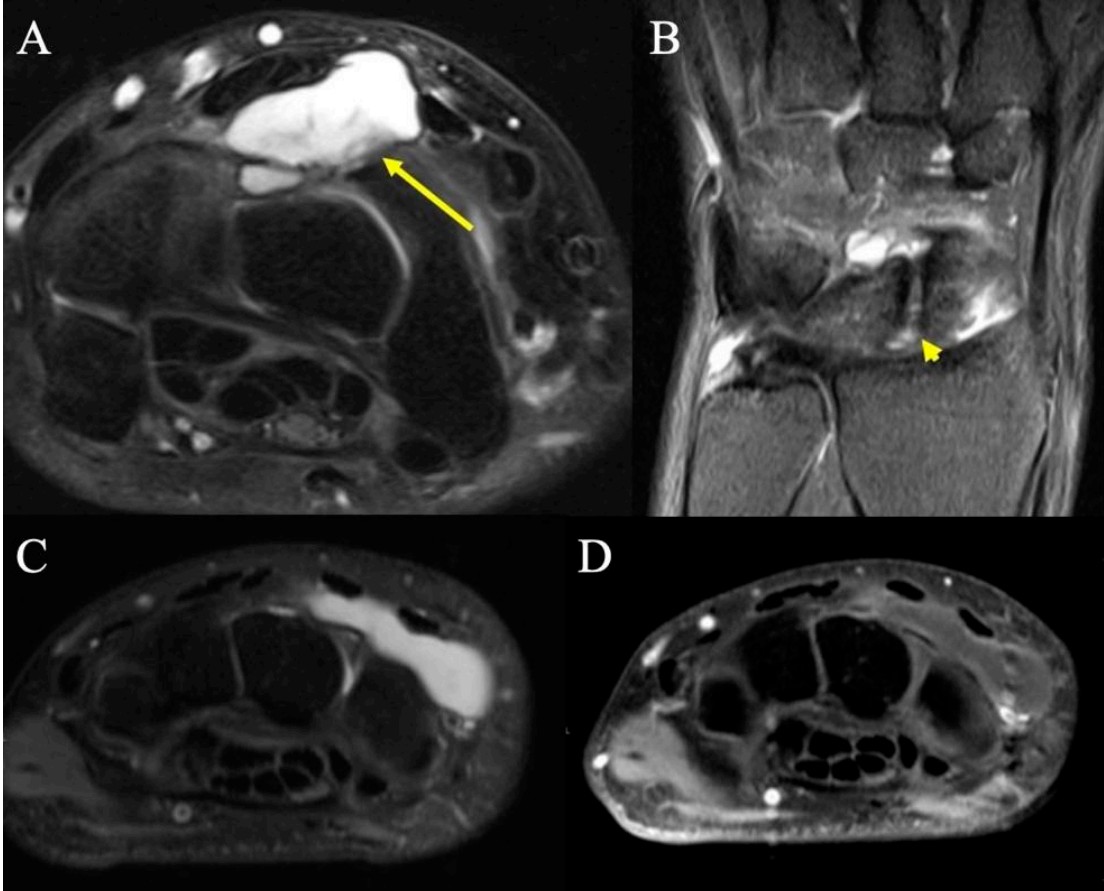

**Figure 1.** Dorsal ganglion. (**A**) Axial fat-suppressed PD MRI shows a moderate-sized dorsal wrist ganglion (arrow) at the proximal carpal row, between the second and fourth extensor compartments, with large component superficial and a smaller component deep to the dorsal intercarpal ligament. (**B**) Coronal fat-suppressed PD shows the neck of the ganglion arising from small rents (arrowhead) in the dorsal scapholunate ligament. (**C**) T2-weighted MR image showing a dorsal homogenous mass deep to the extensor tendons but superficial to the carpal bones at the level of the wrist, consistent with dorsal ganglion. (**D**) Same dorsal hand mass on a T2-fluid attenuated sequence that appropriately loses signal intensity, consistent with dorsal ganglion.

2.1.2. Imaging Appearance

Radiographic and advanced imaging are rarely indicated as the diagnosis of ganglion cysts is typically a clinical one. When obtained, plain radiographs may show associated soft-tissue swelling, but are generally unremarkable. MRI typically shows unilocular or multilocular T2 hyperintense cysts, and careful scrutiny of the adjacent joint capsule often reveals a pathognomonic communicating neck. If symptomatic, wrist ganglia are often amenable to ultrasound-guided needle fenestration and aspiration, though recurrence rates have been reported to be as high as 33–70% [10–12]. Aspiration with the injection of steroid does not significantly alter the recurrence rate [12,13]. Although not truly ganglion cysts, mucoid cysts of the distal interphalangeal (DIP) joints are common (Figure 3) and are associated with osteoarthritis and marginal osteophytes (Heberden's nodes).

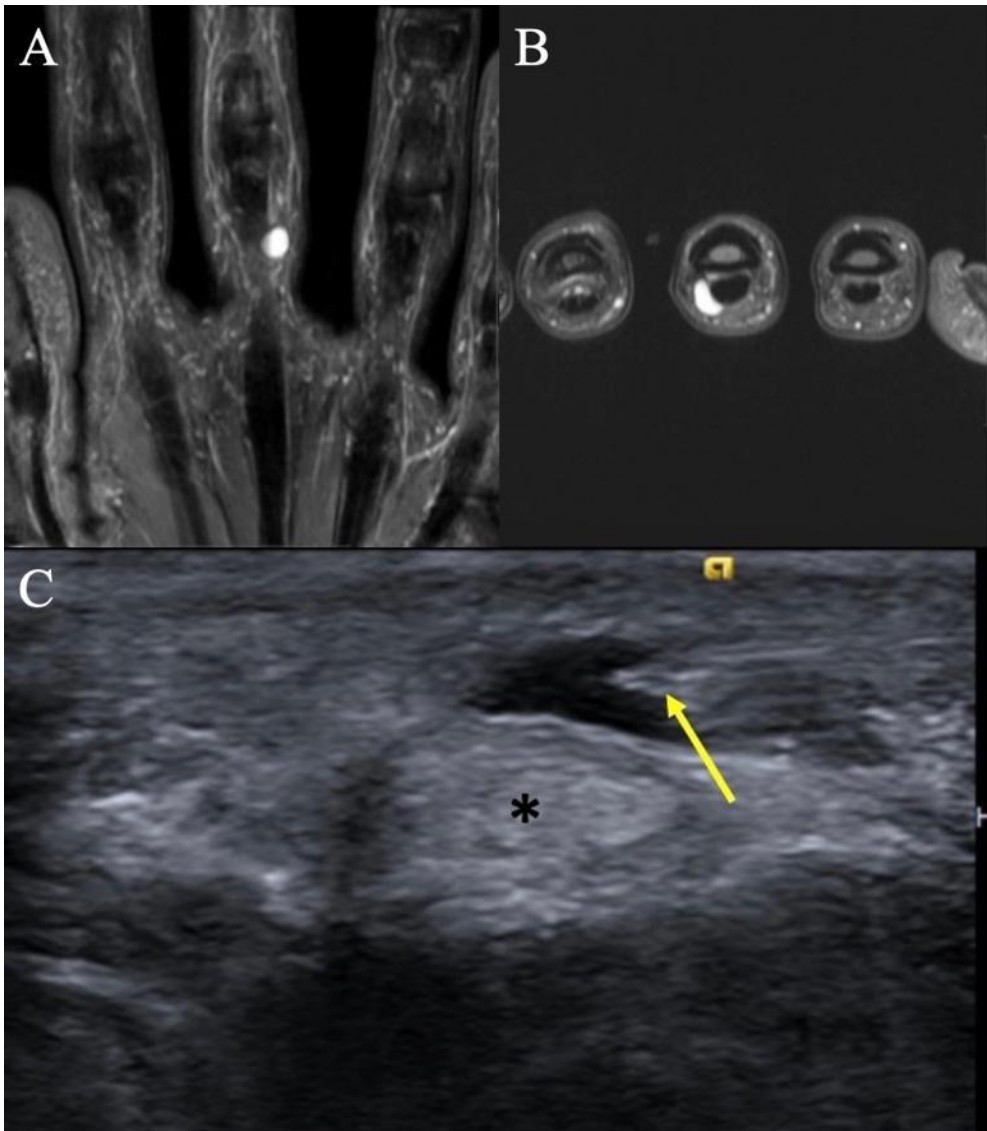

**Figure 2.** Retinacular cyst. (**A**) Coronal T2-weighted image of the hand showing a homogenous, hyperintense mass on the ulnar aspect of the long finger with no internal septations. (**B**) Axial T2-weighted image re-demonstrating the homogenous, hyperintense lesion abutting the flexor tendon. (**C**) Transverse ultrasound at the palmar aspect of the small finger MCP joint in a different patient demonstrates a medial approach for an ultrasound-guided percutaneous needle decompression (arrow) of an A1 pulley retinacular cyst; note safe distance of the needle tip from the flexor tendon (*).

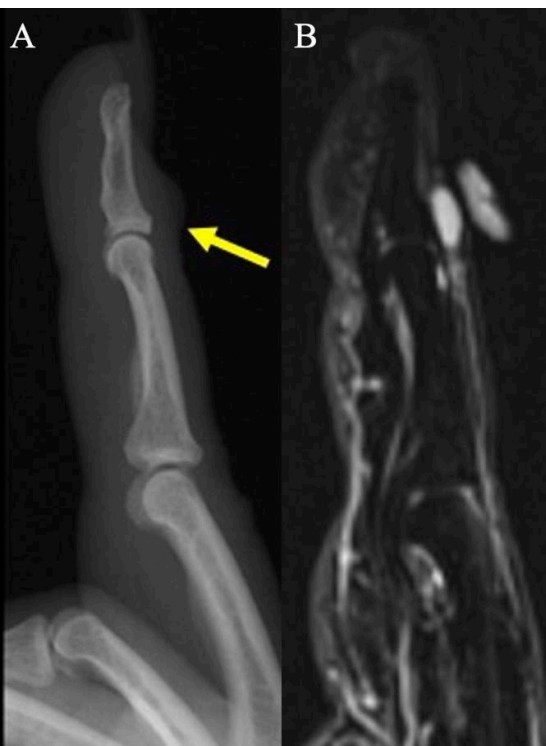

**Figure 3.** Digital mucus cyst. (**A**) Lateral radiograph of index finger demonstrates focal soft-tissue swelling dorsal to the DIP joint, just proximal to the nail fold. (**B**) Sagittal fat-suppressed PD MRI shows a small uniformly hyperintense cyst, in keeping with a digital mucus cyst, without subungual extension or osseous erosion.

### 2.2. Tenosynovial Giant Cell Tumors

#### 2.2.1. Clinical Features

The second most common masses of the hand are tenosynovial giant-cell tumors (TGCTs), historically also known as giant-cell tumors of tendon sheath, or pigmented villonodular synovitis (PVNS) when intra-articular. The global incidence is estimated to be around 43 cases per one million people [14]. These tumors affect patients between the age of 30 and 50 years old, with the median age around 47 years [14,15]. Classically, they present as a single nodule (localized) or as multiple nodules diffusely spread along the tendon sheath, or with pain and swelling of the joint when intra-articular. The most common location is the volar surface of the radial three digits near the distal interphalangeal joints of these fingers [16].

TGCTs are neoplasms, characterized by rearrangements of CSF1 which cause overexpression of macrophage colony-stimulating factor (CSF1) [17] This drives tumor growth, recruitment of non-neoplastic mononuclear and multinucleated inflammatory cells, and benign proliferation of the synovium in the tendon sheath. While non-metastasizing, these tumors may behave aggressively, and recent development of targeted anti-CSF1 agents has enabled systemic treatment options in addition to surgical resection [18].

#### 2.2.2. Imaging Appearance

Radiographs may show erosion of adjacent bones or even joint destruction depending on the size and duration of presence of the lesion. However, 95% of patients present with radiographs that may only show soft-tissue swelling [15].

Advanced imaging is helpful in delineating the extent of disease and severity of adjacent bone or articular damage. Ultrasound typically depicts a heterogeneous hypoechoic mass with internal vascularity on Doppler interrogation. Dynamic ultrasound shows no movement of the mass during passive movement of the surrounding tendons. If localized,

MRI reveals a heterogeneously T1 and T2 hypointense mass that enhances after intravenous contrast administration [19]. These tumors are often associated with intralesional hemorrhage and hemosiderin deposition; because hemosiderin includes paramagnetic $Fe^{+3}$ atoms, the local magnetic field around areas of hemosiderin deposition become distorted, resulting in "blooming" artifacts on gradient-echo or susceptibility-weighted sequences (Figure 4). This effect may be particularly pronounced in diffuse intra-articular forms of disease.

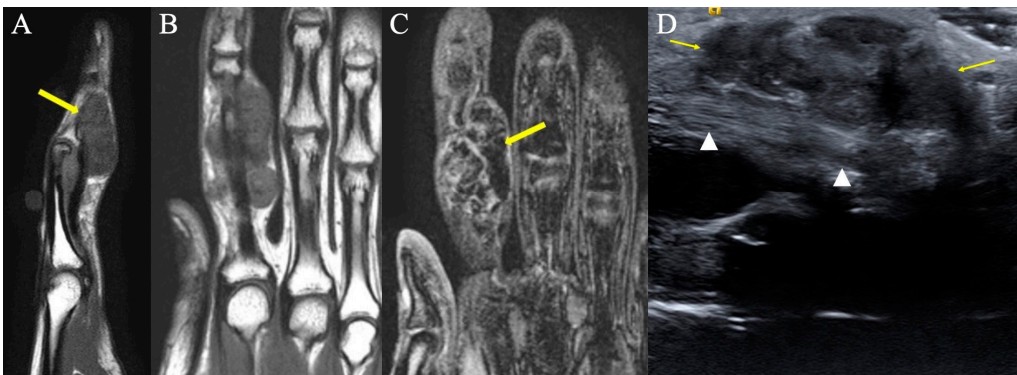

**Figure 4.** Representative images of tenosynovial giant-cell tumors of tendon sheath, previously known as PVNS. (**A**) Sagittal T1-weighted image showing a volar hypointense homogenous mass at the level of the PIP joint. (**B**) Coronal T1-weighted image demonstrating a hypointense homogenous mass on the ulnar aspect of the index finger, consistent with TGCT of tendon sheath. (**C**) Gradient-echo coronal image of the hand demonstrating a hypointense mass of the index finger with associated distorted artifact signal (yellow arrow), so called "blooming" artifacts consistent with TGCT of tendon sheath. This characteristic appearance is secondary to intralesional hemorrhage and hemosiderin deposition; because hemosiderin includes paramagnetic $Fe^{+3}$ atoms, the local magnetic field around areas of hemosiderin deposition become distorted. (**D**) Longitudinal ultrasound of the index finger shows the palmar component of the tenosynovial giant-cell tumor as a hypoechoic lobular mass (arrows), volar to the flexor tendon (arrowheads). The diagnosis was proven by ultrasound-guided percutaneous needle biopsy.

### *2.3. Lipoma*

#### 2.3.1. Clinical Features

Although the most common soft-tissue tumor of the body, lipomas of the hand are relatively uncommon, accounting for 8% of benign tumors of the hand [20,21]. Adipocytic tumors are commonly associated with translocations and rearrangements of the 12q13-15 chromosomal region [22]. It is believed that these abnormal karyotypes lead to chromosomal fusion products which promote proliferation of adipocytes. They also result in the upregulation of MDM2, an oncogenic signature of atypical lipomatous tumor (ALT)/well-differentiated liposarcoma, with ALT being the preferred term for tumors in the extremities. Biologic transformation of ALT to dedifferentiated liposarcoma arises rarely but only after further genomic instability and generation of neochromosomes [23].

Lipomas of the hand present as slowly enlarging, soft non-tender masses. Symptoms of compression neuropathy can be present if the lesion is in the vicinity of peripheral nerves (e.g., carpal tunnel or Guyon's canal). On examination, a mobile mass with a characteristic "doughy" texture is the classic description. To differentiate from ganglion cysts, lipomas do not transilluminate. A positive Tinel's sign over the tumor can confirm adjacent nerve compression as the source of paresthesia.

#### 2.3.2. Radiographic Appearance

On plain radiographs, an area of radiolucency within the soft tissues may be noted, and smooth scalloping of adjacent bones may be observed. Ultrasound examination will reveal a well-circumscribed lesion that is homogenous and often hyperechoic in nature,

although lipoma echogenicity is variable [24]. Magnetic resonance imaging will show a mass that has the same signal characteristics of subcutaneous fat tissue (Figure 5). The mass may be well-demarcated with a fibrous pseudocapsule, but can also be poorly marginated, rendering it occult on imaging. Typically, low-grade lipomatous neoplasms do not enhance with contrast, although the fibrous pseudocapsule may show mild enhancement.

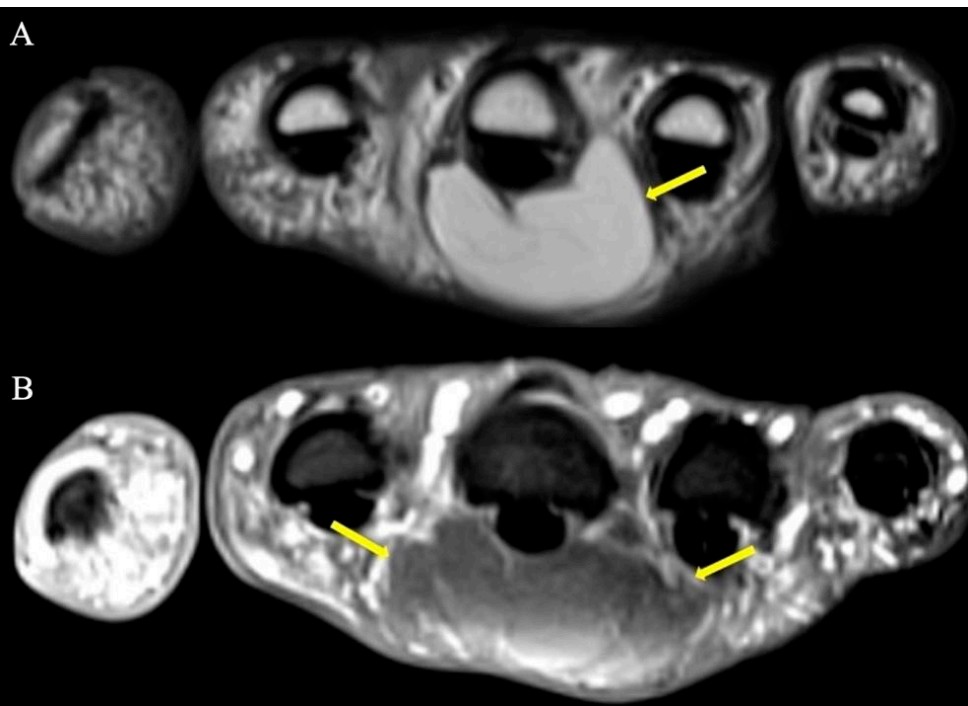

**Figure 5.** Palmar lipoma hand-simulating retinacular cyst. (**A**) T1 and (**B**) fat-suppressed T2-weighted axial images demonstrating a lipoma (arrows) situated just volar to the 3rd metacarpal and associated flexor tendon. Note the homogenous appearance and similar signal characteristics to subcutaneous fat on both pulse sequences.

*2.4. Schwannoma*

2.4.1. Clinical Features

These are the most common benign tumors of peripheral nerves, making up about 5% of all benign soft-tissue neoplasms [25]. They are derived from Schwann cells and myelin sheaths and are well-encapsulated and slow-growing [26]. These tumors rarely involve the nerves of the hand and as benign tumors, malignant transformation is exceptional [27]. These tumors can present at any age, with an increased frequency particularly within the third and sixth decades of life, and there is no gender predilection [26]. These tumors often present as a painful mass, often with Tinel sign, and with careful excision, the peripheral nerve can be spared.

2.4.2. Radiographic Appearance

Schwannomas show low-signal intensity on T1W images and may display a "target sign" appearance on fluid-sensitive sequences with central hypointensity and peripheral hyperintensity, owing to collagenous and myxomatous components, respectively (Figure 6) [28]. While schwannomas are eccentrically located allowing excision with nerve salvage, they are often not radiologically distinguishable from neurofibromas [29]. MRI also aids in surgical planning to ensure optimum nerve recovery and minimize unnecessary nerve damage during resection of the tumor.

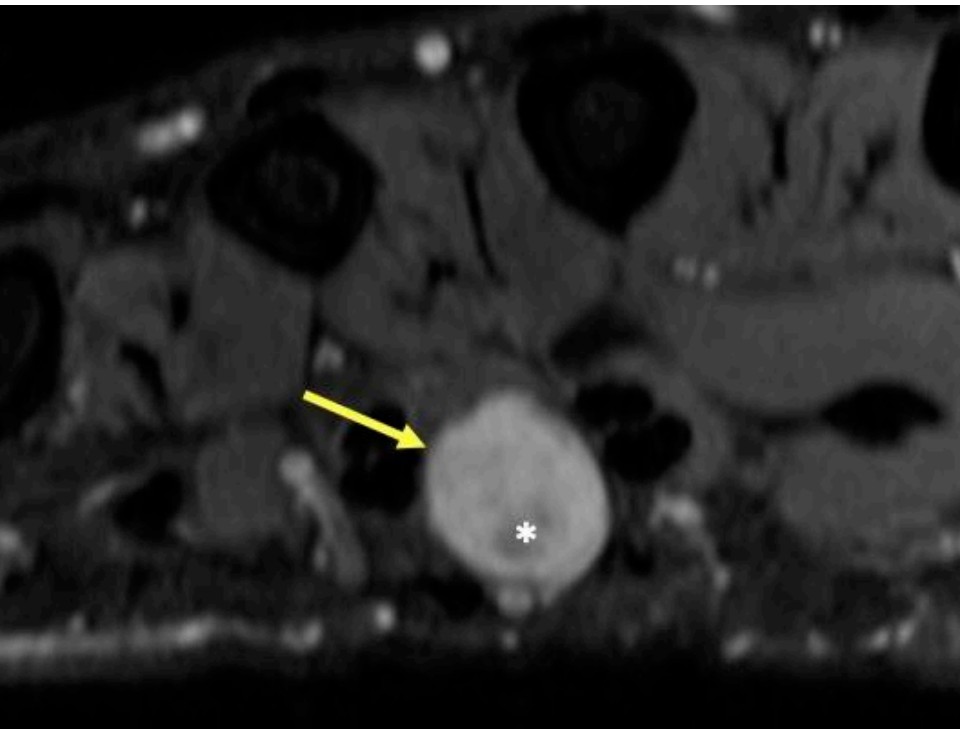

**Figure 6.** Patient with neurofibromatosis type II and mass in the third webspace of the hand. T2-weighted axial image showing a round hyperintense mass (arrow) with a central region of low signal intensity (*), characteristic of peripheral nerve sheath tumor, in this case a schwannoma of the common digital nerve. The hypointense central portion in T2-weighted images is attributable to a dense area of collagenous stroma; when this collagenous region is larger, the mass may assume a "target sign" appearance.

### 2.5. Glomus Tumors

### 2.5.1. Clinical Features

Glomus tumors are defined as perivascular neoplasms of the glomus body of the distal phalanx and subungual region. Most are benign, but some can be malignant and demonstrate aggressive behavior. The glomus body of the finger is a neuroarterial structure responsible for thermoregulation and is located underneath the nails or in the pulp of the finger. Accounting for about 1–5% of tumors in the hand, they generally present between the fourth and sixth decade of life [30]. Solitary glomus tumors are slightly more common in females than males. The etiology of many glomus tumors is idiopathic, although up to half harbor NOTCH-gene fusions, and BRAF mutations in a smaller subset [31,32].

Subungual and phalangeal glomus tumors tend to be less than 1 cm in size and present with the classic triad of symptoms: local sensitivity, pain with cold exposure, and severe pain following minor trauma [33]. Because of its rarity and miniscule size, the typical time from onset of symptoms to a correct diagnosis is around seven years [34]. Subungual involvement can lead to discoloration or deformation of the nail plate.

### 2.5.2. Imaging Appearance

Plain radiographic imaging of a glomus tumor is usually unremarkable. Rarely, cortical scalloping of the underlying phalanx can be found in long-standing cases. Ultrasound studies typically show hypoechogenicity, internal vascularity on Doppler interrogation, and tenderness to direct sonographic palpation with the ultrasound prob [35,36]. On MRI, glomus tumors manifest as smooth-contoured lesions, with low or intermediate signal on T1W images, with hyperintensity on T2W images (Figure 7) [37]. Because of the vascularity of the lesions, these lesions display characteristic uniform and avid contrast enhancement.

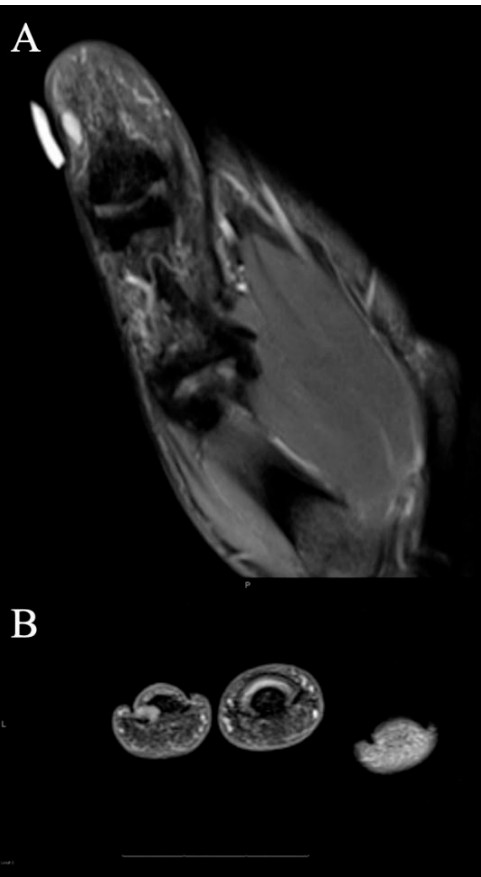

**Figure 7.** Glomus tumor. (**A**) Coronal fat-suppressed PD MRI shows a small 5 mm T2 hyperintense mass abutting the radial cortex of the thumb distal phalanx, consistent with glomus tumor that was subsequently excised. Note good correlation with overlying marker indicating site of pain. (**B**) T2-weighted axial image at the level of the paronychial folds that show a hyperintense mass just adjacent to the distal phalanx. The mass was excised and found to be a glomus tumor.

### 2.6. Vascular Tumors and Malformations

2.6.1. Clinical Features

Low-flow malformations include venous malformations (VMs), capillary malformations (CMs), and lymphatic or venolymphatic malformations (LMs or VLMs) [38]. Arteriovenous malformations (AVMs) are the classic type of high-flow lesions that are commonly a result of prior trauma or invasive procedures within the hand. Historically in the radiology literature and in clinical practice, it is common to see the term "hemangioma" used to describe venous malformations.

The most common presentation of VMs is that of a soft-tissue mass that is painful with activity or cold weather. On examination, these soft-tissue masses usually have a bluish tint and increase in size with dependent positioning, while decompressing when the upper extremity is elevated due to changes in hydrostatic pressure. The lesions may cause compression of nearby nerves and corresponding paresthesia, pain, or weakness.

LMs have a classic rubber texture on palpation and can have overlying skin dimpling. Distinction from VMs can be made because LMs do not change in size based on the position of the upper extremity. Radiologic distinction can be difficult, particularly as some entities contain components of both as a venolymphatic malformation. CMs present as red or pink cutaneous discolorations that are commonly referred to as port-wine stains. Finally, AVMs are often warm, pulsatile masses that have associated bruits and thrills on examination due to arterialized blood flow through engorged veins. A proximal AVM may result in symptoms of digital ischemia and finger discoloration. Similar to LMs and CMs, AVMs do not change in size based on the position of the upper extremity

### 2.6.2. Imaging Appearance

Plain radiographs of low-flow VMs may show phleboliths, pathognomonic calcifications within thrombosed veins, in the area of suspicion. These phleboliths will be hyperechoic on ultrasound, or can result in signal void on MRI (Figure 8). Sonographic imaging demonstrates a hypoechoic mass with monophasic low-velocity flow.

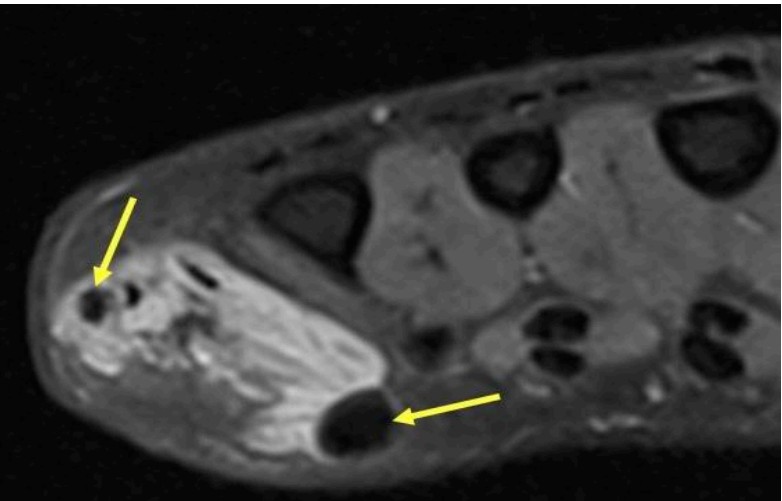

**Figure 8.** Venous Malformation. Axial T2-weighted axial image showing a hyperintense irregular mass on the ulnar aspect of the hand with interspersed hypointense foci (yellow arrows) consistent with phleboliths in a venous malformation. The round hypointense phleboliths are pathognomonic calcifications within thrombosed veins, and ideally would be corroborated with radiography (though unavailable in this case).

Finally, ultrasound and color Doppler ultrasound can reliably assess the pulse flow characteristics of an AVM. CT angiography and venography is helpful in delineating the enlarged veins with arterialized flow (Figure 9). MRI imaging demonstrates tubular structures with flow voids on both T1W and T2W imaging [39]. Time-resolved MR angiography may be of use to help demonstrate early-draining veins, outline the location of shunts, and delineate the extent of collateral vasculature.

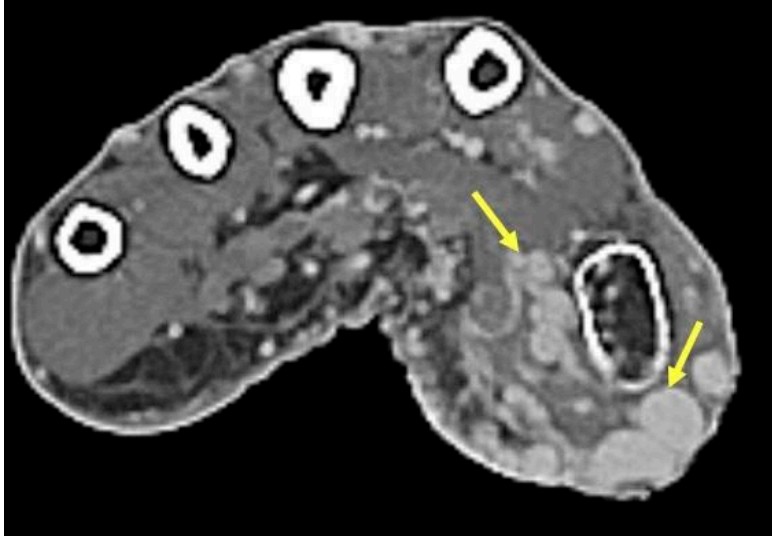

**Figure 9.** Arteriovenous malformation. Axial contrast-enhanced CT demonstrates numerous engorged vessels in the thenar eminence without a discrete mass in this high-flow arteriovenous malformation.

### 2.7. Superficial Fibromatoses

#### 2.7.1. Clinical Features

Superficial palmar fibromatosis of the hand, also known as Dupuytren's disease, is a connective-tissue disorder that results in firm, subcutaneous nodules on the palmar surface of the hand that can cause debilitating flexion contractures. Known as the "Disease of the Vikings", there is a strong predilection towards Caucasian males of northern European descent, with an incidence of roughly 30 per 100,000 annually [35]. In general, men are affected more than women at a ratio of about 2:142, and hereditary forms have been described [35].

Characteristic nodules and cords along the palmar fascia are thought to result from dysregulated fibroblastic proliferation mediated by fibrogenic cytokines: fibroblast growth factor, wingless/integrated (Wnt), and transforming growth factor β have all been implicated [36,40]. Disease most commonly affects the ring and small finger, but can affect any digit of the hand, typically starting in the palm and progressing distally [41]. Depending on the severity of disease, there may be overlying skin dimpling and pitting.

#### 2.7.2. Imaging Appearance

Plain radiographs generally add little value in the imaging of palmar fibromatosis. On ultrasound, hypoechoic nodules that arise from the palmar fascia are characteristic. MRI reveals low-signal intensity both on T1W and T2W images, a common theme of lesions that are of composed of dense collagen. Intermediate signal intensity on T2W MRI suggests higher cellularity and a higher tendency towards recurrence [42]. Recent MRI studies have shown that T2 mapping can be used to quantify the extent of lesion collagenization in response to treatment and maturation [43] (Figure 10).

### 2.8. Synovial Chondromatosis

#### 2.8.1. Clinical Features

Synovial chondromatosis is a rare benign neoplasm that generally affects patients in their fourth through sixth decades [44]. The disease more commonly occurs within the larger joints of the knee, hip, and shoulder. Pain, swelling, and decreased range of motion of the affected joint are some of the clinical symptoms. Recently, gene rearrangements involving FN1 and/or ACVR2A were detected in a majority of cases [45].

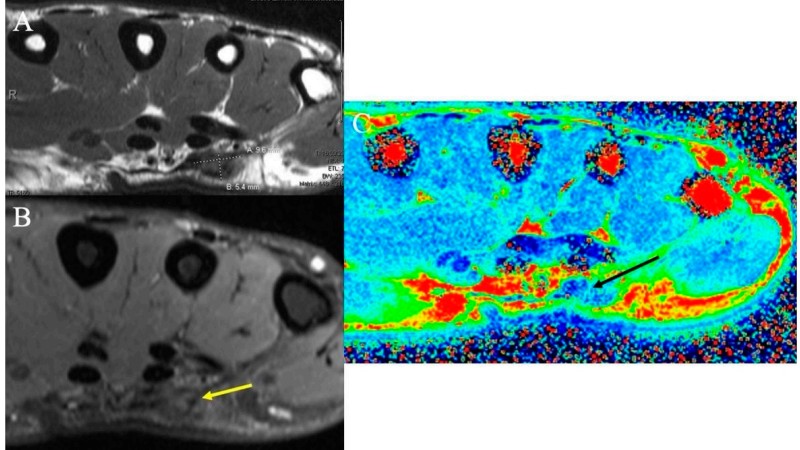

**Figure 10.** Dupuytren's disease. (**A**) Axial T1-weighted MR image demonstrating a low-signal mass measuring 9.6 mm × 5.4 mm consistent with the diagnosis of superficial palmar fibromatosis or Dupuytren's disease. (**B**) Axial fat-suppressed PD-weighted MR image demonstrating the low-signal nodule (yellow arrow); high collagen content within the nodule results in the relative hypointensity of superficial fibromatosis. (**C**) Axial color-coded T2 map MRI demonstrates the nodule (black arrow) as reflecting tissue with decreased T2 relaxation times (blue color), the characteristic of collagen-rich fibromatosis.

### 2.8.2. Imaging Appearance

Some 70–95% of synovial chondromatosis cases display multiple intra-articular calcifications [44]. The "ring-and-arc" mineralization pattern recapitulates the lobular morphology of enchondral ossification observed in cartilaginous tumors. Synovial chondromatosis may result in scalloping and extrinsic erosion of bone on both sides of the involved joint, usually best depicted on CT or MRI. On T2-weighted MRI, noncalcified areas of neoplastic hyaline cartilage show high intensity [45] (Figure 11).

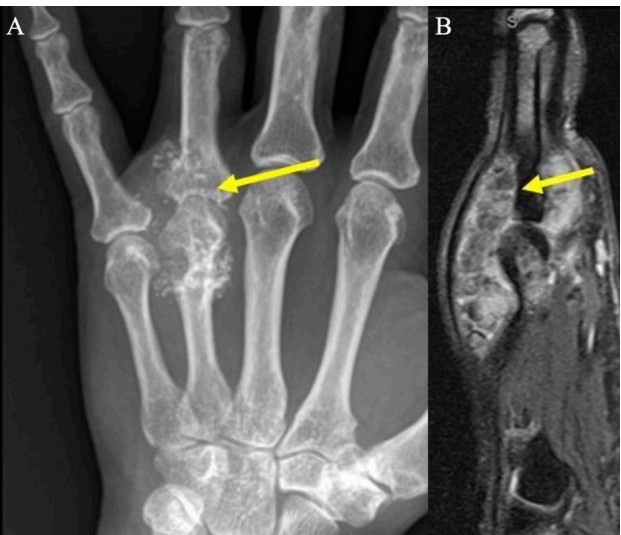

**Figure 11.** Chondromatosis. (**A**) AP radiograph shows numerous calcified masses with chondroid pattern mineralization at the fourth metacarpophalangeal joint (arrow), with smooth scalloping and extrinsic erosions of the adjacent metacarpal neck and proximal phalanx. (**B**) Sagittal fat-suppressed PD demonstrates numerous hypointense chondral bodies (arrows) along both the flexor and extensor tendons, scalloping the bone, consistent with synovial chondromatosis.

### 2.9. Infectious Tenosynovitis

#### 2.9.1. Clinical Features

Tenosynovitis is the inflammation of the tendon sheath and is an orthopedic emergency when caused by infection [46]. Because of its structure as a fluid within a closed compartment, the tendon sheath is susceptible to significant damage in the setting of increased compartment pressures. Other more common causes include inflammation and trauma. The most common location of infectious tenosynovitis is the flexor tendon sheaths as a result of trauma in the skin [47].

Infectious tenosynovitis can lead to irreversible disability and morbidity due to the resulting contracture of involved musculature. Common microorganisms in order of prevalence include Staphylococcus aureus, Pasteurella multocida (cat bites), Neisseria gonorrhoeae (sexually transmitted), Eikenella corrodens (human bites), and Mycobacteria tuberculosis [48].

#### 2.9.2. Imaging Appearance

Due to its higher-contrast resolution, MRI is the most sensitive and specific imaging modality for the diagnosis of infectious tenosynovitis [46]. Fluid accumulation within the tendon sheaths and synovial hyperplasia can be detected on MRI. Especially in tenosynovitis caused by Mycobacteria tuberculosis, a nonspecific "rice-bodies" appearance may be seen. With gadolinium on contrast-enhanced MRI, the paratendinous contrast enhancement caused by diffuse inflammation can be demonstrated. In addition, CT with contrast demonstrates tendon thickening, edema, and abscess.

### 2.10. Foreign Bodies of the Hand

### 2.10.1. Clinical Features

Because hands are frequently used in daily activities, they are vulnerable to penetration and impalement injuries that result in foreign bodies in the hand. Foreign bodies may remain asymptomatic if initially overlooked, but over time may lead to inflammatory, allergic, and infectious complications [49]. Complications include foreign-body granulomas, pyogenic granulomas, abscess, chronic discharging wound, and impingement on neighboring bones, tendons, blood vessels, and nerves. Prevention of these complications consists of surgical removal.

### 2.10.2. Imaging Appearance

The index of suspicion for a foreign body should be high in any penetrating hand injury and warrants imaging investigation especially given a suggestive history [50]. Plain radiographs with multiple views should be the first-line imaging investigation. Ultrasound can detect non-radiopaque foreign bodies and assist in foreign-body removal (Figure 12). If a suspected foreign body is not found in other imaging modalities, CT and/or MRI may be helpful in identifying small wood or wood-like fragments and delineate the associated soft-tissue inflammation and other foreign-body complications.

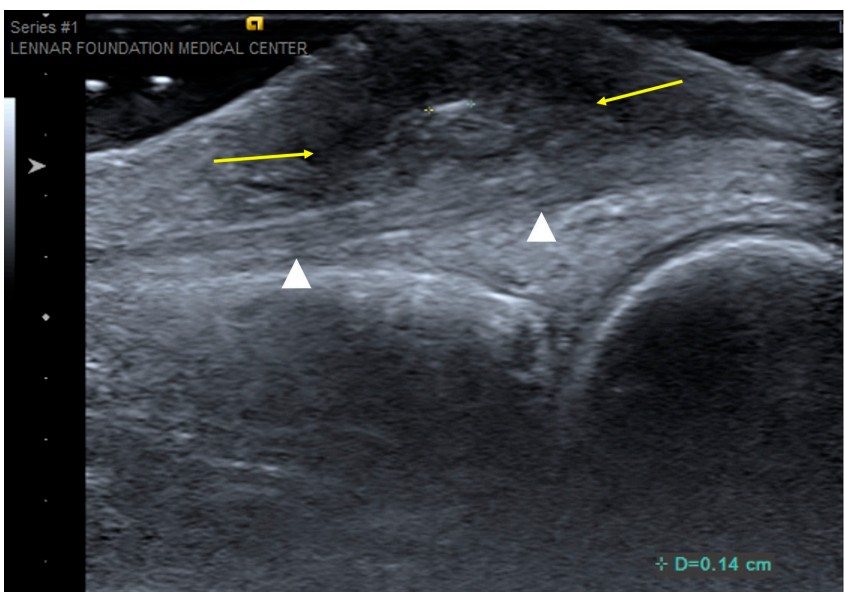

**Figure 12.** Foreign-body granuloma. Longitudinal ultrasound over dorsal aspect of the middle finger at the level of the metacarpophalangeal joint shows a 1.4 mm echogenic foreign body (digital calipers) and surrounding heterogeneously hypoechoic inflammatory soft-tissue reaction (arrows). The underlying extensor tendon (arrowheads) was confirmed to be normal at surgery.

## 3. Soft-Tissue Malignancies

### 3.1. Clinical Features

Of all soft-tissue sarcomas (STS), only about 20% originate in the upper extremity [2]. These soft-tissue tumors arise from extraskeletal mesenchymal tissues. In the pediatric population, fibrosarcomas and rhabdomyosarcomas are the most common hand soft-tissue sarcomas.

Epithelioid sarcomas, synovial sarcomas (Figure 13), and undifferentiated pleomorphic sarcomas represent the most common type of malignant soft-tissue sarcomas of the hand in adults [51]. Acral myxoinflammatory fibroblastic tumor is a lower-grade sarcoma with a predilection for the hand and fingers (Figure 14). Typically, STS presents as a painless growing mass; patients may attribute incidental trauma to the area.

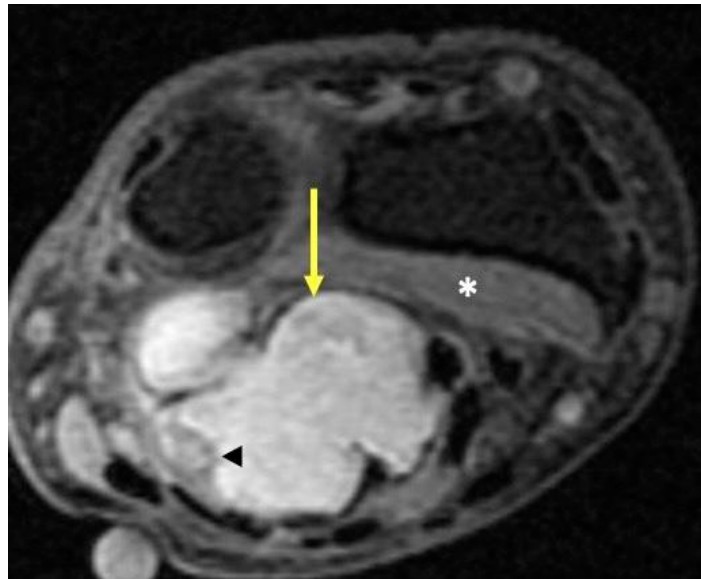

**Figure 13.** Synovial sarcoma. Fat-suppressed T2-weighted axial MR image of the wrist just proximal to the distal radioulnar joint shows a high-signal-intensity lobulated volar mass (arrow) deep to the superficial flexor tendons and overlying the pronator quadratus (*). Internal areas of necrosis and hemorrhage may lead to areas of signal heterogeneity due to evolving blood products within the tumor (arrowhead).

In the event of a diagnosis of a soft-tissue sarcoma, MRI is required to map anatomic involvement of the neuro-vasculature, tendons, and bone so that resectability can be determined, and appropriate reconstructive options can be planned. Typically, STS have low signal on T1W images and are heterogeneously high-signal intensity on T2 sequences. In higher-grade STS or carcinosarcoma, a non-enhancing necrotic center with peripheral enhancing zone of viable tumor may be observed (Figure 15), which can lead to mischaracterization as an abscess. Depending on tumor size and grade, appropriate staging studies need to be performed, which may include lymphoscintigraphy for those STS with a propensity to metastasize to nodes (e.g., epithelioid sarcoma).

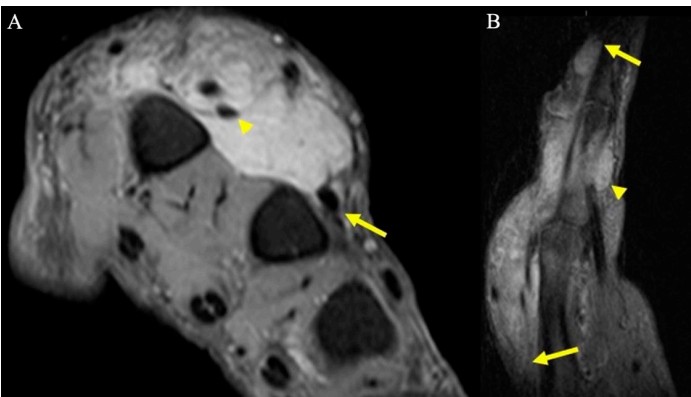

**Figure 14.** A 67-year-old man with soft tissue mass of the hand. (**A**) Axial fat-suppressed PD MRI shows a large hyperintense soft-tissue mass in the dorsum of the hand, centered over the second and third metacarpals, and encasing the index finger extensor tendons (arrowhead), and thin tails of infiltrating tumor partially encasing middle finger extensor tendon (arrow). (**B**) Sagittal fat-suppressed PD MRI highlights the extent of the infiltrative mass along the index finger dorsally (arrows), as well as smaller volar components (arrowhead). Initially, this was thought to be an abscess but was later revealed to be acral myxoinflammatory fibroblastic sarcoma after incisional biopsy, and ultimately required second and third ray amputations to achieve wide negative margins.

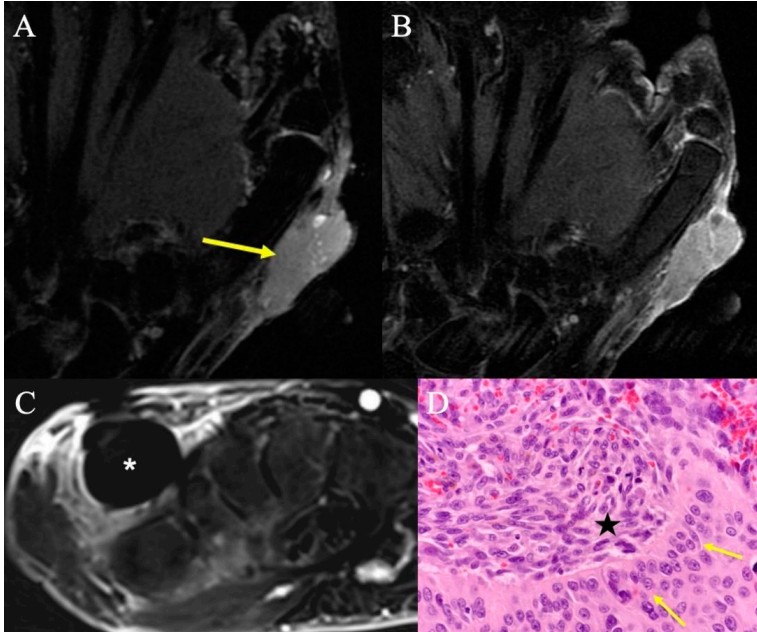

**Figure 15.** Carcinosarcoma. (**A**) Coronal fat-suppressed T2-weighted image showing a homogenous mass nearly isointense to muscle, radial to the first metacarpal (arrow). (**B**) Coronal contrast-enhanced fat-suppressed T1-weighted imaging shows mildly heterogeneous enhancement of this solid mass that involves the skin and subcutis. Biopsy confirmed carcinosarcoma of the hand. (**C**) Contrast-enhanced fat-suppressed T1-weighted image with subtraction from routine surveillance MRI 12 months after surgical resection revealed a 1.5 cm round mass (*) in the dorsal aspect of the surgical resection bed; despite lack of central enhancement, the morphology and timing of the imaging findings raised concern for recurrent disease, which was subsequently proven by percutaneous needle biopsy. (**D**) High-powered H&E showing the sharp demarcation between the squamous cell component (arrows) and spindle-shaped cells (*) characteristic of carcinosarcoma of the hand.

*3.2. Surgical Considerations*

STS of the hand and fingers present unique challenges as input from multiple surgical subspecialists may be required, including orthopaedic oncology, hand surgery, and plastic surgery, in discussions with patients for goals of care and outcome expectations. The need to achieve wide negative tumor margins often necessitates radical resection with reconstruction of tendon, bone, nerve, and skin in pursuit of complete tumor extirpation, although ray amputation may ultimately be required for limb salvage [52,53]. Multidisciplinary planning with hand and plastic surgery is paramount in successful restoration of hand function and soft tissue coverage. Planned positive margins with adjuvant radiation may be acceptable in certain situations where local recurrence risk is outweighed by preservation of limb function [54], although outcomes in such cases are usually poor [55].

## 4. Discussion

Soft-tissue tumors of the hand are relatively commonly encountered in clinical practice. Although the vast majority of these lesions are benign, they may generate a great deal of anxiety for patients. Knowledge of the more common hand soft-tissue tumors, understanding their presentations, and recognizing their imaging characteristics can help radiologists and clinicians appropriately treat and advise patients. In this review, the complexity of the various soft-tissue tumors of the hand was synthesized and discussed. The basic epidemiology, clinical presentation, imaging findings, and treatment options of each tumor were highlighted. The overarching goal of the paper is to educate trainees, provide a reference to practicing physicians, and spread awareness about this topic.

**Author Contributions:** Conceptualization, S.A., M.P.R., F.F.S. and T.S.; methodology, S.A., M.P.R., F.F.S. and T.S.; investigation, S.A., M.P.R., F.F.S. and T.S; resources, S.A., M.P.R., F.F.S., D.D., A.R., J.P., N.F. and T.S.; writing—original draft preparation, S.A., M.P.R., F.F.S. and T.S.; writing—review and editing, S.A., M.P.R., F.F.S., D.D., A.R., J.P., N.F. and T.S.; visualization, S.A., M.P.R., F.F.S. and T.S.; supervision, T.S.; project administration, T.S. All authors have read and agreed to the published version of the manuscript.

**Funding:** This research received no external funding.

**Institutional Review Board Statement:** Ethical review and approval were waived because this project is not human subject research requiring IRB review, approval or oversight. This determination is based on the definition of "human subject" found at 45 CFR 36.102(e). This research does not involve interactions with living human subjects or accessing identifiable information or identifiable biospecimen, which does not meet the definition of human subject research.

**Informed Consent Statement:** Patient consent was waived because this project is not human subject research and does not involve identifiable private information or identifiable biospecimens.

**Conflicts of Interest:** The authors declare no conflict of interest.

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
