# Peer review of "Soft Tissue Masses of the Hand: A Review of Clinical Presentation and Imaging Features"

_curroncol, doi:10.3390/curroncol30020158_

Round 1

Reviewer 1 Report

I appreciate the manuscript and I suggest some improvements/changes:

- Discussion section is completely missing.  Please, provide it.

- Please, check keywords according to MESH terms. In particular, check if 'benign' and 'malignant' used alone are MESH terms, I don't think so..

- Please, check all figure legends. Indicate correctly all the MRI sequences shown (e.g. in Figures 1 and 2 the T2w sequences are fat suppressed)

- I suggest You substitute the term Radiological with 'Imaging' a more comprehensive term that better also includes ultrasound

- In the section focused on benign conditions, I suggest you to add a common entity in the hand (foreing bodies) that may present acutely or later as a lump. Moreover, infectious tenosynovitis with its most common location in hand and wrist in the flexor tendons, should be added too. (suggested updated reference Spinnato P, Patel DB, Di Carlo M, Bartoloni A, Cevolani L, Matcuk GR, Crombé A. Imaging of Musculoskeletal Soft-Tissue Infections in Clinical Practice: A Comprehensive Updated Review. Microorganisms. 2022; 10(12):2329. https://doi.org/10.3390/microorganisms10122329)

- Moreover, I suggest focusing a bit more deepen in regards to the differential diagnosis among cysts and GCT of tendons sheets - An image of GCT would be highly appreciated if possible (especially US appearance, useful in clinical practice)

Author Response

Thank you for taking the time to review our manuscript. We added the discussion section and revised our keywords so that they are according to MESH terms. We adjusted figure legends and substituted the term "radiological" to "imaging" to be more comprehensive.  We added infectious tenosynovitis and foreign bodies of the hand. Lastly, we added an ultrasound image of GCT (Figure 4D). 

Reviewer 2 Report

A well written concise overview of hand masses, very educational for trainees or a reference for practicing physicians.

The Introduction should not be a word for word copy of the Abstract. The introduction should give more relevant detail to what is to come, or provide other overarching facts that tie hand tumors together.

51 - lining

86 - tenosynovial

131 - Need reference for this data

158 - Add "B)" to figure

250 - Per earlier statement and more recent convention, consider calling this a VM instead of hemangioma.

285) "B" also looks like an axial, not sagittal, sequence. In general the Figure description of Figure 10 does not appear to match the images presented.

Author Response

Thank you for taking the time to review our manuscript. We added an Introduction section that is different than the abstract. We also made the revisions suggested for lines 51, 86, 131, 158, 250, and 285. We fixed the description for Figure 10. 

Reviewer 3 Report

The topic of this reports is review radiologic an clinical features of  soft masses  detected in  the hand. Overall the topic is interesting and may have clinical relevance for readers. Nevertheless, there are some minor points which need to be arranged prior to consider this manuscript for publication.

1. Although discussion in this paper is not required, a brief summary or conclusion should be included.

2. References section should be reviewed. There are typographical mistakes  (example, ref. 9),  information is missed in some references (example ref.35) or use the acronyms of the journals should be properly used (example, ref. 13).

Author Response

Thank you for taking the time to review our manuscript. We have added a discussion section and revised the references section accordingly.

Reviewer 4 Report

The work looks more like a book chapter rather than a Review.

I suggest introducing and discuss the latest literature sources otherwise it is the equivalent of a book chapter.

The pictures are interesting.

Many works cited are between 2000 and 2010, more than 10 years ago.

Author Response

Thank you for taking the time to review our manuscript. We have added more recent literature sources. 

Round 2

Reviewer 1 Report

I appreciated greatly the revisions performed, thank you.

Reviewer 2 Report

Thank you for revisions

Reviewer 3 Report

All the queries have been corrected. Thank you. Therefore, I consider this manuscript to be published in this journal.

Reviewer 4 Report

The authors have improved their work significantly.